# Comparative Study on In Vitro Fermentation Characteristics of the Polysaccharides Extracted from Bergamot and Fermented Bergamot

**DOI:** 10.3390/foods12152878

**Published:** 2023-07-28

**Authors:** Kaizhang Wu, Xingyu Lu, Fang Fang, Juncheng Liu, Jie Gao, Yang Zheng

**Affiliations:** 1Chaozhou Branch of Chemistry and Chemical Engineering Guangdong Laboratory, Chaozhou 521031, China; 2216401019@st.gxu.edu.cn; 2Department of Food Science, School of Light Industry and Food Engineering, Guangxi University, Nanning 530004, China; 2016301019@st.gxu.edu.cn (X.L.); 2016391007@st.gxu.edu.cn (F.F.); liu_juncheng2006@st.gxu.edu.cn (J.L.)

**Keywords:** bergamot, polysaccharide, in vitro fermentation, gut microbiota, metabolome

## Abstract

This study aimed to compare the in vitro fermentation characteristics of polysaccharides from Bergamot and Laoxianghuang (fermented 1, 3, and 5 years from bergamot) using the stable in vitro human gut fermentation model. Results showed that bergamot polysaccharide (BP) and Laoxianghuang polysaccharides (LPs) with different surface topographies were characterized as mannorhamnan (comprising Mannose and Rhamnose) and polygalacturonic acid (comprising Galacturonic acid and Galactose), respectively. The distinct effects on the gut microbiota and metabolome of BP and LPs may be due to their different monosaccharide compositions and surface morphologies. BP decreased harmful *Fusobacterium* and promoted beneficial *Bifidobacterium,* which was positively correlated with health-enhancing metabolites such as acetic acid, propionic acid, and pyridoxamine. *Lactobacillus*, increased by LPs, was positively correlated with 4-Hydroxybenzaldehyde, acetic acid, and butyric acid. Overall, this study elucidated gut microbiota and the metabolome regulatory discrepancies of BP and LPs, potentially contributing to their development as prebiotics in healthy foods.

## 1. Introduction

In recent years, the risk of diet-related chronic diseases has increased in the human population, which is closely linked to intestinal health [1]. The human intestine, which contains microorganisms with a population of up to 10^14^, is a complex system and provides a good environment for microorganisms to inhabit [2]. Gut microbiota are an essential bridge between diet and human health, and stabilize human metabolic health to prevent common human metabolic disorders [3]. Dietary polysaccharides are attracting more and more attention because of their various biological activities, and play a role by interacting with gut microbiota, which contribute to promoting both beneficial bacteria growth and short-chain fatty acids (SCFAs) production [4,5]. Furthermore, certain polysaccharides such as inulin are considered prebiotics, which as a dietary supplement are not digested and absorbed in gastrointestinal tract, but are utilized by gut microbiota to promote host health [6].

Bergamot (*Citrus medica* L. var. *sarcodactylis Swingle*), classified in the family Rutaceae and the genus Citrus, is mainly planted in Guangdong, Fujian, Yunnan, Sichuan, and other provinces in China. Bergamot is widely regarded for its high medicinal and therapeutic value in Chinese medicine and food-based products. Bergamot contains many active substances, such as flavonoids, hesperidin, polysaccharides, alkaloids, and coumarins [7,8]. A previous study reported that bergamot-derived dietary fiber reduced the ratio of *Firmicutes* to *Bacteroides* in order to optimize gut microbiota composition in rats [9]. Laoxianghuang is also utilized as a traditional medicinal plant, obtained from bergamot through salting, desalting, sugaring, cooking, drying, and fermentation (pickle and storage for different time periods) [10]. In our previous work, bergamot polysaccharides (BP) and Laoxianghuang polysaccharides (LPs) were determined to regulate potential imbalances in the gut microbiota and metabolome in patients with hyperlipidemia; there were notably distinct structure characterizations: (1) BP consists mainly of 63.62% Mannose and 24.16% Rhamnose. (2) LPs contain main monosaccharide types such as Galacturonic acid and Galactose [11]. However, the information is limited in the relationship between the structure and effects of BP and LPs on the human gut microbiota and metabolome.

Therefore, our work aimed to investigate the effects of polysaccharides from BP and LPs with different structures on the human gut microbiota using a stable in vitro human gut fermentation model created by our previous study, which was effective in keeping a stable bacterial community structure in the human gut during in vitro fermentation [12]. The preliminary structural characterizations of BP and LPs were evaluated via Fourier transform infrared spectroscopy (FT-IR) and scanning electron microscopy (SEM). 16S rRNA, GC-MS, and UPLC-MS/MS were used to analyze the effects of polysaccharides on the composition of gut microbiota, the production of SCFAs, and the non-targeted metabolomics, respectively. The results provided information on the effects and differences of BP and LPs as potential prebiotics in the regulation of gut microbiota and metabolites.

## 2. Materials and Methods

### 2.1. Materials and Reagents

Fresh bergamot fruits and Laoxianghuang were purchased from the local market in Guangzhou, Guangdong Province, China. Standards of SCFAs were purchased from Shanghai Aladdin Biochemical Technology Co., Ltd. (Shanghai, China), including formic, acetic, propionic, butyric, i-butyric, valeric, and i-valeric acid. Ethyl acetate of HPLC grade was supplied by Chengdu Kelong Chemical Co., Ltd. (Sichuan, China), and methanol of HPLC grade was supplied by Tianjin Shield Special Chemical Co., Ltd. (Tianjin, China). Acetonitrile of HPLC grade was supplied by Thermo Fisher Scientific Co., Ltd. (Shanghai, China). All other used reagents were analytical grade.

### 2.2. Preparation of the Polysaccharides

BP were isolated from fresh bergamot fruits as previously described with some modifications [13]. Fresh bergamot fruits were used as raw materials, the bergamot fruit skins were removed, and 100 g of bergamot fruits were accurately weighed. After drying at 40 °C, the weighed bergamot fruits were homogenized in a crusher (2500A, Red Sun Co., Ltd., Yongkang, China) for 10 min to form a powder. The powder was transferred to a 2000 mL beaker and 1000 mL of deionized water was added. The stirring device was installed and placed in a water bath at a constant temperature of 80 °C for 5 h to extract BP polysaccharides. The supernatant was collected and concentrated under a vacuum using a rotary evaporator (RE-52AA, Yarong biochemical instrument factory, Shanghai, China) at 60 °C. The concentrated extract was precipitated using ethanol (95%, *w*/*v*) to an ethanol concentration of 80% (*w*/*v*) to extract polysaccharides for 8 h. The precipitates were collected after centrifugation at 6876× *g* for 15 min. The precipitants were mixed with trichloroacetic acid (TCA) to reach a final concentration of 4% (*w*/*v*), and placed at 4 °C for 10 h. After centrifugation at 6876× *g* for 15 min, the precipitates were removed. The supernatant was collected and dialyzed (molecular weight cut-off: 8000–14,000 Da) to remove the TCA for 48 h and lyophilized to obtain BP. LPs were extracted in the same method. Different LPs at 1, 3, and 5 years of fermentation were obtained, which were coded as MLP, TLP, and PLP, respectively.

### 2.3. Structural Characterization

Fourier transform infrared spectroscopy (FT-IR), scanning electron microscopy (SEM) and monosaccharide composition analysis were conducted. The functional groups of four polysaccharides were identified using an FT-IR spectrophotometer (TENSOR II, BRUKER, Mannheim, Germany) within 4000–400 cm^−1^. Approximately 2 mg of sample and 100 mg of dried potassium bromide (KBr) were mixed and ground in an agate mortar. The mixture was pressed into a sheet for FT-IR analysis. The microcosmic morphology of the four polysaccharides were observed using a SEM (Phenom ProX, Phenom-World, Utrecht, Netherlands). The lyophilized sample of the four polysaccharides were sputtered with gold. Micrographs recorded the appropriate magnification to obtain clear images at an acceleration voltage of 10.0 kV. The monosaccharide composition was determined according to our previous study [11].

### 2.4. In Vitro Fermentation of Polysaccharides

The fermentation of polysaccharides was carried out with the stable in vitro model for human gut microbiota fermentation according to our previous work [12]. This model was effective in keeping a stable bacterial community structure and simulating the nutrient-poor microenvironment in the human gut during in vitro fermentation. In brief, the fresh media of the in vitro human gut fermentation model was mixed with 0.5% of different polysaccharides and cultured in an anaerobic workstation (HYQX-III-Z, Shanghai Yuejin Medical Instrument Co., Ltd., Shanghai, China) for 48 h, which were defined respectively as BP, MLP, TLP, and PLP groups. Three independent sample replicates were performed for each group. Three samples from each group were collected separately at 0, 12, 24, 36, and 48 h after fermentation. They were stored at −80 °C for further study.

### 2.5. DNA Extraction and 16S rRNA Gene Sequencing

The genomic DNA was extracted using the HiPure stool DNA kits (Magen, Guangzhou, China) based on the manufacturer’s instructions. PCR amplicons were created from the target region V3-V4, forward primer (314F: CCTACGGGNGGCWGCAG), and reverse primer (806R: GGACTACHVGG-GTATCTAAT). Raw data containing adapters or low-quality reads would affect the following assembly and analysis. Raw reads were further filtered to obtain quality reads by using FASTP [14] (version 0.18.0). The original sequence and the paired terminal reads were merged into tags by quickly adjusting the length of short fragments with FLASH [15] (version 1.2.11). The noise sequence of the original tag was filtered under specific filtering conditions to obtain high-quality clean tags. A UPARSE (version 9.2.64) pipeline was used to cluster clean labels into operational taxonomic units (OTUs) with a similarity of ≥97%. All chimeric tags were removed using the UCHIME algorithm [16]. Effective tags were eventually obtained for further analysis. The most abundant labeled sequence was selected as the representative sequence in each cluster.

### 2.6. Determination of SCFAs

The measurement of the SCFAs in each in vitro fermented sample was carried out according to our previous method with slight modifications [17]. Briefly, 500 μL of sample was mixed with 5 μL of formic acid and placed at −20 °C for 2 h. Following that, samples were centrifuged at 4 °C and 15,300× *g* for 10 min. A total of 400 μL of supernatant was mixed with 400 μL of ethyl acetate for 2 min. Subsequently, the mixture was centrifuged at 4 °C and 4352× *g* for 10 min. The supernatant of each mixture was dried with sodium sulfate and filtered through a 0.22 μm organic phase filtration membrane. The pretreated samples were injected into a GC-MS system equipped with GC (Agilent 7890B mag Agilent Technologies, Santa Clara, CA, USA), highly inert MSD (Agilent 5977A mag Agilent Technologies, Santa Clara, CA, USA), and a TG-WAXMS column (60 m × 0.25 mm × 0.25 µm, Thermo Science, Waltham, MA, USA). The injection temperature was 250 °C, the injection volume was 1 μL with automatic injection, the split ratio was 1:1, and the flow rate of helium was 1 mL/min. The heating procedure was to increase the temperature from 90 °C to 150 °C at a rate of 15 °C/min, and then raise the temperature to 170 °C at a rate of 5 °C/min. The temperature was eventually raised to 220 °C at a rate of 20 °C/min and maintained for 2 min.

### 2.7. Metabolomics Analysis

For the extraction and analysis of metabolites, we followed the method previously reported, with some modifications [18]. Fermented sample (100 μL) was added to 600 μL of MeOH and vortexed for 60 s. Then, the mixture was centrifuged for 15 min at 11,492× *g* and the supernatant was obtained. The step regarding methanol extraction and centrifugation was repeated twice. Subsequently, the supernatant of two centrifugations was mixed and used for high-performance liquid chromatography−tandem mass spectrometry (HPLC−MS/MS) analysis. HPLC−MS/MS analysis was conducted on a Q Exactive Plus system (Thermo-fisher Scientific, Waltham, MA, USA), coupled online to an HPLC system, and equipped with a Hypersil GOLD C18 column (100 × 2.1 mm,1.9 um. Thermo-fisher Scientific). The column temperature was 30 °C, and the flow rate was 0.3 mL/min. The mobile phase consisted of A: 0.1% formic acid aqueous solution and B: acetonitrile. A gradient-elution program was performed and set as follows: Starting with 5% B for 2 min, the linear gradient was increased to 95% B over 14 min, followed by 5% B isocratic elution for 2 min. Positive and negative ion modes of electrospray ionization (ESI) were used for metabolite detection at 3.0 kV. The data were collected according to the range (100 to 1000 *m/z*) under a resolution of 70,000 FWHM. For dd-MS2, the mass spectra were recorded at a resolution of 17,500 FWHM, with an AGC value of 1 × 105 within 50 ms. Fragment ions were generated in HCD collision cells using stepped normalized collision energy (NCE 10, 25, and 45%). Compound Discoverer software 3.2 (Thermo Scientific, Waltham, MA, USA) was performed to analyze the results.

### 2.8. Statistical Analysis

The data were presented as mean ± SD (n = 3) and evaluated by one-way analysis of variance (ANOVA), followed by Tukey’s post hoc test using IBM SPSS statistics 26.0 software. GraphPad Prism 8.0 and Origin 2018 software were used for the graphic presentation.

## 3. Results

### 3.1. Structural Characterization

FT-IR spectroscopy is an effective tool for characterizing polysaccharide functional groups. As shown in Figure 1, the FT-IR spectra of the LPs were clearly different from BP. They showed typical absorption peaks of polysaccharides at around 3400, 2930, and 1030 cm^−1^, which were attributed to the stretching vibration of the O-H, C-H, and C-O bonds, respectively. The characteristic absorption peak at 1740 cm^−1^ and 1620 cm^−1^ indicated the stretching vibration of the ester carbonyl (C=O) and asymmetric stretching of the carboxylate anions (COO-), respectively. At 1410 cm^−1^, the absorption peak was ascribed to symmetric stretching of COO- in BP and LPs. In BP, the peak at 2850 cm^−1^ was attributed to the symmetric stretching vibrations of CH_2_. The weak bands at 890 cm^−1^ may be attributed to the β-glycosidic linkage for BP. Furthermore, the characteristic bands between 1000 and 1200 cm^−1^ might correspond to ring vibrations, with stretching vibration overlap of the C-OH side groups and the C-O-C glycosidic bond vibration, representing the presence of pyranose sugars. The SEM images of BP and LPs were shown in Figure 2. The results showed that the different fermentation periods induced different physical changes in size and shape. The surface of BP was smooth and demonstrated relatively regular and homogeneous shapes (Figure 2A). However, the surface of MLP was rough, showing web-like layers of varying sizes (Figure 2B). At higher magnitudes of 3000×, TLP was observed to have irregular protruding particles (Figure 2C). The irregular trait with a rougher surface was displayed in PLP (Figure 2D). Polysaccharide surface morphology could be affected by the fermentation time. The results of monosaccharide compositions analysis showed that BP was composed mainly of Mannose and Rhamnose, which was characterized as mannorhamnan. However, LPs mainly consisted of Galacturonic acid and Galactose, which were characterized as polygalacturonic acid (Appendix A) [11].

### 3.2. Effect of Polysaccharides on Gut Microbiota

To elucidate the effects of BP, MLP, TLP, and PLP on the microbiota profile throughout the in vitro fermentation, we assessed the bacterial composition for samples collected from 48 h through multiplex sequencing covering the V3-V4 regions of the 16S rRNA. The Shannon–Wiener rarefaction curves showed that no additional OTUs were detected with the increase in sequencing depth and sample size (Appendix A). The outcome suggested that the data from the gut microbial community were robust. The Chao1 and the Simpson indices were used to evaluate the difference in the diversity of gut microbiota among the 4 groups. LPs resulted in a significantly decreased diversity in gut microbiota compared to the control group (*p* < 0.05) (Appendix A). To study the changes in the gut microbiota community composition after fermentation in vitro, principal coordinates analysis (PCoA) was carried out at the OTU level based on Bray–Curtis distance. PCo1 and PCo2 contributed 37.83% and 27.94% of the variation, respectively, and PCoA revealed that the microbiota shifted after the intervention of BP and LPs (Figure 3A). Appendix A showed the relative abundance of gut microbiota at the phylum level. *Proteobacteria*, *Firmicutes*, and *Bacteroidetes* were the dominant bacteria, and the ratio of *Firmicutes* to *Bacteroidetes* was significantly declined compared with the control group (Figure 3B, *p* < 0.05). Gut microbiota composition was also significantly affected by the polysaccharide intervention at genus level (Figure 3C). As shown in Figure 3D, a remarkable increase of *Bacteroides* and *Faecalibacterium* was observed in polysaccharide-treated groups (BP and LPs), while a significant decrease was exhibited for *Tyzzerella*, *Peptoclostridium*, and *Lachnoclostridium* in BP and LPs (*p* < 0.05). In addition, BP and MLP markedly reduced the relative abundance of *Fusobacterium*, and TLP and PLP elevated *Megasphaera*. (*p* < 0.05). BP, TLP, and PLP significantly enhanced the growth of *Bifidobacterium* (*p* < 0.05). LPs dramatically contributed to increasing the relative abundance of *Lactobacillus* (*p* < 0.05).

### 3.3. SCFAs Production

The effects of different polysaccharides on SCFAs production was evaluated during fermentation (0, 12, 24, 36, and 48 h). After fermentation, the SCFAs levels gradually increased in all groups from 0 to 48 h. Acetic, propionic, and butyric acids were the major components. The acetic acid contents were the highest at 48 h (Appendix A). Compared to the control group, all treatments showed significant increments of total SCFA production after 48 h (Figure 4A, *p* < 0.05). In BP treatment, the content of total SCFAs and both acetic and propionic acid was remarkably higher than the LPs groups, reaching 57.26 ± 0.14, 20.76 ± 0.18, and 17.97 ± 0.21 mmol/L, respectively (Figure 4A–C, *p* < 0.05). The content of butyric acid in the MLP group was notably higher than the other groups, up to 13.01 ± 0.08 mmol/L (Figure 4D, *p* < 0.05). From a comprehensive perspective, BP could promote more SCFAs production than LPs. In LPs, MLP facilitated production of the highest yield of butyric acid.

### 3.4. Effects of Polysaccharides on Metabolic Profiles

To further explore the potential effects of BP and LPs on intestinal metabolites, untargeted metabolomics was performed among different groups. Principal component analysis (PCA) was conducted to exhibit the metabolic distinction. The PCA scores plot showed a clear divergence differentiation in the control, BP, and LPs groups (Figure 5A). Compared to the control group, polysaccharides-supplemented groups showed significant alteration by fold change (FC, *p* < 0.05), and the number of upregulated (FC > 2) metabolites was higher than the downregulated metabolites (FC < 0.5) (Figure 5B). In more detail, a total of 168 metabolites was markedly regulated in the BP group (136 upregulated and 32 downregulated). There were 310 (199 upregulated and 111 downregulated), 349 (235 upregulated and 114 downregulated), and 356 (241 upregulated and 115 downregulated) for MLP, TLP, and PLP groups, respectively.

Variable importance projection (VIP) values are often used to identify significant markers contributing to differences among groups. We selected the top 20 metabolites for further study within the VIP values (Figure 6A). The content of pyridoxamine was elevated in BP group (Appendix A). There was an upward trend in 4-Hydroxybenzaldehyde under the influence of LPs (Appendix A). Spearman’s correlation analysis between the gut microbiota at the genus level and the metabolites (top 20 within the VIP values and SCFAs) was implemented to elucidate the association between gut microbiota and metabolites (Figure 6B). The relative abundance of *Bacteroides* and *Erysipelotrichaceae_UCG-003* showed a positive correlation with acetic acid (*p* < 0.05). *Faecalibacterium*, *Lactobacillus*, and *Erysipelotrichaceae_UCG-003* were discovered to be correlated positively with butyric acid (*p* < 0.05). *Peptoclostridium* and *Phascolarctobacterium* were found to be positively correlated with propionic acid (*p* < 0.05). The relative abundance of *Fusobacterium* and *Escherichia-Shigella* exhibited a significant negative correlation with pyridoxamine (*p* < 0.05). *Lactobacillus* showed positive correlation with butyric acid and 4-Hydroxybenzaldehyde (*p* < 0.05).

## 4. Discussion

In recent years, prebiotics modulating gut microbiota to enhance human health have attracted great attention [3]. Many polysaccharides as macromolecular prebiotics have been confirmed to stimulate the growth of probiotics in the intestine and improve the host’s metabolism [19]. The previous study suggested that *Hericium erinaceus* polysaccharides promoted SCFA-producing bacteria and beneficial bacteria, and inhibited opportunistic pathogenic bacteria [20]. Ai et al. found that black mulberry fruit polysaccharides could be utilized by human gut microbiota. High levels of SCFAs were produced during polysaccharide fermentation [21]. *Dictyophora indusiata* polysaccharides and okra pectic-polysaccharides have the potential to regulate gut microbiota and improve intestinal health [22,23]. However, few studies report the effects and differences of Bergamot and Laoxianghuang polysaccharides on gut microbiota composition and metabolism in healthy individuals. In our work, we extracted the crude polysaccharides of bergamot and Laoxianghuang (fermented for 1, 3, and 5 years), and compared their structural differences. An in vitro fermentation was performed to compare the gut microbiota and metabolite differences among various polysaccharides.

Our previous results found that the monosaccharide composition of LPs was more complex than that of BP. This was similar to the previous study, in that the number of monosaccharide types in polysaccharides increased as the storage period of Chenpi was prolonged [24]. Here, FT-IR analysis revealed that BP and LPs had characteristic functional groups of polysaccharides (O-H, C-H, and C-O) [25]. The absorption peaks at 1740 cm^−1^ and 1620 cm^−1^ showed the presence of C=O and asymmetric stretching of COO- [26]. The symmetric stretching of COO- in BP and LPs were revealed at 1410 cm^−1^ [27]. BP exhibited differences in CH_2_ and β-glycosidic linkage stretching vibration compared to LPs [28,29]. Pyranose sugars may be present in BP and LPs [30]. SEM analysis demonstrated that the surface morphology of BPs and LPs were dramatically different. The surface of BP was smoother than LPs, which may be beneficial when being touched and utilized by gut microbiota.

The advantage of the in vitro model is that most of the disturbances from the complex environment are eliminated. In our research, we found that LPs markedly decreased the microbial diversity, probably because insufficient carbohydrates in the fermentation system inhibited the growth of gut microbiota [31]. It is commonly believed that obese individuals exhibit a high ratio of *Firmicutes* to *Bacteroidetes* [32]. BP and LPs reduced the ratio of *Firmicutes* to *Bacteroidetes*, suggesting that BP and LPs may contribute to the treatment of obesity. Furthermore, BP and LPs impacted the growth of certain genera which may be prospectively beneficial for maintaining host health, such as *Bacteroides*, *Faecalibacterium*, *Bifidobacterium*, *Megasphaera*, and *Lactobacillus*. However, MLP had no effects on the growth of *Bifidobacterium*. *Lactobacillus* was not affected by BP. *Bacteroides* have been reported to improve intestinal inflammation and are considered to be key players in cancer immunotherapy and prevention [33]. *Bifidobacterium* and *Lactobacillus* are recognized as probiotics and have anti-inflammation and anti-cancer effects [34]. Recently, the research found that *Megasphaera* might play a critical role in improving the host lipid metabolism [35]. On the other hand, BP and LPs suppressed the reproduction of harmful bacteria. *Fusobacterium* was significantly downregulated in BP and MLP groups. TLP and PLP cannot down-regulate the relative abundance of *Fusobacterium*. The relative abundance of *Tyzzerella*, *Peptoclostridium*, and *Lachnoclostridium* was diminished. *Fusobacterium* may be an essential pathogen, which induces colorectal cancer [36]. Significantly increased *Tyzzerella* was found in patients with chronic inflammatory bowel disease [37]. *Peptoclostridium* gave rise to intestinal infection and disease symptoms ranging from mild diarrhea to life-threatening pseudomembranous colitis [38]. *Lachnoclostridium* was shown to produce harmful lipid compounds such as trimethylamine and cytidine diphosphate diacylglycerol, and cause negative implications for host metabolic health [39]. Thus, BP and LPs promoted some potentially beneficial microbial populations and suppressed some potentially harmful bacterial growth. BP and LPs may have different effects on gut microbiota because of differences in the types of monosaccharide composition and surface morphology [40].

SCFAs are important products of polysaccharide degradation by gut microbiota, which contribute positively to the improvement of intestinal health [41]. SCFAs levels, such as acetic, propionic, and butyric acids, gradually increased in all groups. The phenomenon may be explained by a significant increase in the *Bacteroides.* Polysaccharides are cleaved into oligosaccharides and monosaccharides by *Bacteroides* to provide the fermentation substrate for other SCFAs-producing bacteria [42]. Different SCFAs content with the same trend in LPs groups may be attributed to a similar type of monosaccharide composition [43]. BP and LPs showed the ability to promote SCFAs production. The highest butyrate concentration was found in the MLP group, which may be related to the enriched *Faecalibacterium*. Our result demonstrated that the concentration of butyric acid was positively correlated with *Faecalibacterium*. This was similar to the previous study, and *Faecalibacterium* is an important butyric-acid-producing bacterium in the human intestine [44]. Therefore, BP and LPs may result in differences in the production of SCFAs due to variations in monosaccharide composition.

The non-target metabolomics approach was conducted to investigate alterations in gut microbial metabolites. Pyridoxamine, a significant marker found among five groups, was upregulated under the effects of BP. Others reported that pyridoxamine might inhibit the synthesis of pro-inflammatory cytokines to protect from liver fibrosis [45] and suppress the upregulation of pro-inflammatory genes in the visceral adipose tissues of HFD mice [46]. The correlation revealed that *Bacteroides*, *Faecalibacterium*, *Bifidobacterium*, and *Lactobacillus* were positively correlated with acetic or butyric acid. Pyridoxamine had a positive correlation with *Bacteroides* and *Bifidobacterium*. *Fusobacterium* was negatively correlated with acetic acid, propionic acid, butyric acid, and pyridoxamine. There was a positive correlation between *Lactobacillus* and 4-Hydroxybenzaldehyde. 4-Hydroxybenzaldehyde had an upward trend with the influence of LPs. A study suggested that 4-Hydroxybenzaldehyde could regulate gut microbiota and have the potential to inhibit lipid accumulation [47]. BP (comprising Mannose and Rhamnose) and LPs (comprising Galacturonic acid and Galactose) may have differences affecting gut microbiota due to their structural discrepancies, thus causing differences in the associated metabolite levels.

## 5. Conclusions

The present study revealed the different modulatory effects of BP and LPs on gut microbiota and metabolite for healthy individuals. Our work displayed that BP and LPs may have distinct effects on the gut microbiota and metabolome due to their different monosaccharide compositions and surface morphologies. BP (characterized as mannorhamnan), mainly consisting of Mannose and Rhamnose, decreased the relative abundance of harmful *Fusobacterium* and promoted beneficial *Bifidobacterium*. Nevertheless, the relative abundance of *Lactobacillus* was increased by LPs (mainly composed of Galacturonic acid and Galactose, and characterized as polygalacturonic acid). BP and LPs as preferable fermentation substrates were utilized by gut microbiota, and increased the SCFAs production. BP may be favorable to be utilized by gut microbiota due to the smooth surface morphology. Furthermore, BP could be metabolized to produce potentially health-beneficial metabolites such as pyridoxamine, which is positively associated with *Bacteroides* and *Bifidobacterium*. 4-Hydroxybenzaldehyde, acetic acid, and butyric acid were positively correlated with *Lactobacillus*, which was up-regulated by LPs. Overall, there are differences in the potential regulation effects of BP and LPs on gut microbiota, SCFAs, and metabolites because of their different structural characteristics. However, follow-up animal experiments are needed to further explore the mechanisms which cause the differing effects of BP and LPs on the gut microbiota and metabolomics of humans. This may contribute to boosting the applications of BP and LPs as potential prebiotics in healthy foods.

## Figures and Tables

**Figure 1 foods-12-02878-f001:**
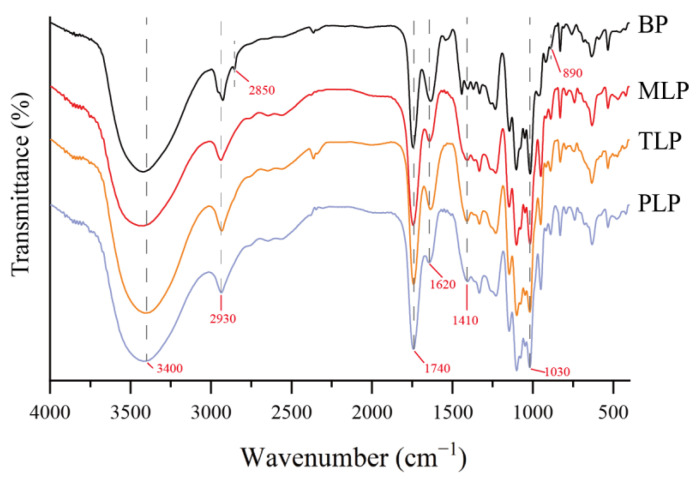
Fourier transform infrared spectroscopy (FT-IR) of four polysaccharides (BP, MLP, TLP, and PLP).

**Figure 2 foods-12-02878-f002:**
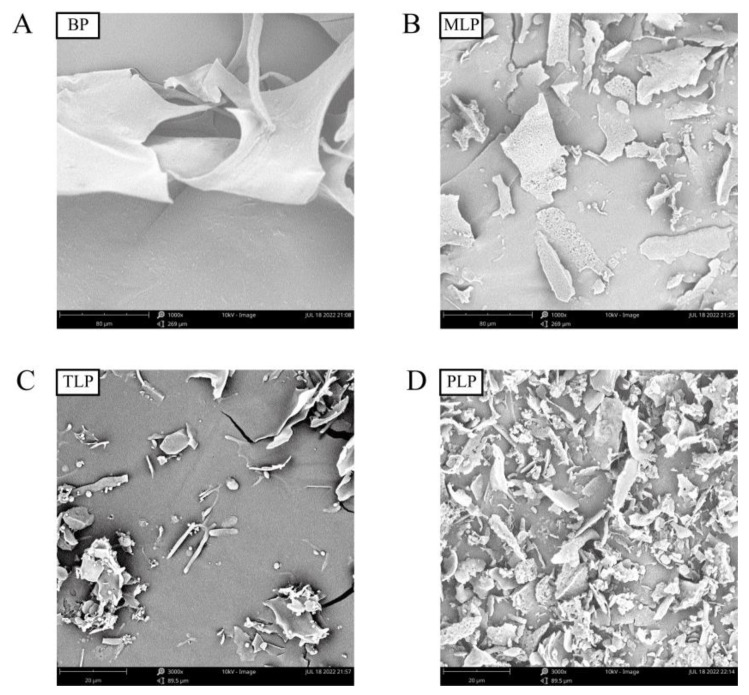
Scanning electron micrographs of four polysaccharides. (**A**) BP (1000×), (**B**) MLP (1000×), (**C**) TLP (3000×), (**D**) PLP (3000×).

**Figure 3 foods-12-02878-f003:**
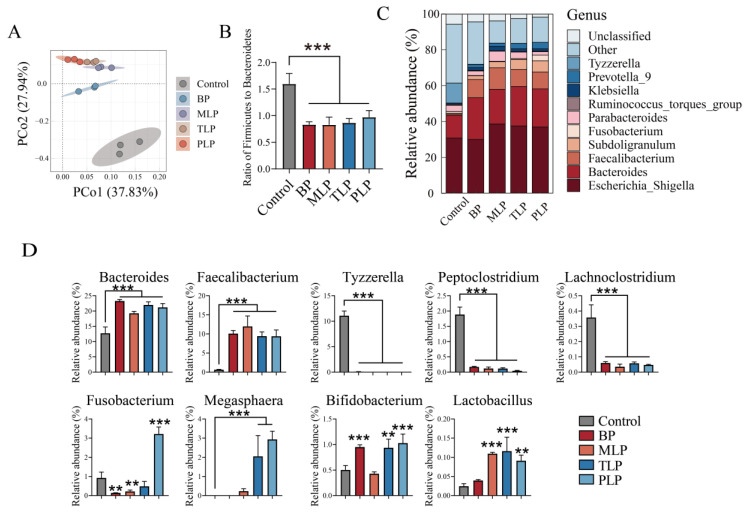
Effect of different polysaccharides on the regulation of gut microbiota after in vitro fermentation. (**A**) Principal coordinates analysis (PCoA) of fecal fermentations, (**B**) the ratio of *Firmicutes* to *Bacteroidetes* in vitro fermentation at 48 h, (**C**) the relative abundance at the genus level in fermentation broth at 48 h from different treatments, (**D**) and the relative abundance of *Bacteroides*, *Bifidobacterium*, *Megasphaera*, *Lactobacillus*, *Peptoclostridium,* and *Lachnoclostridium*. Data were expressed as mean ± SD (n = 3), and evaluated by one-way ANOVA with Tukey’s post hoc test. ** *p* < 0.01, *** *p* < 0.001, compared to the control group. The control group (no additional carbon source supplement); BP (Bergamot polysaccharide supplement); MLP (mono-Laoxianghuang polysaccharide supplement); TLP (tri-Laoxianghuang polysaccharide supplement); PLP (penta-Laoxianghuang polysaccharide supplement).

**Figure 4 foods-12-02878-f004:**
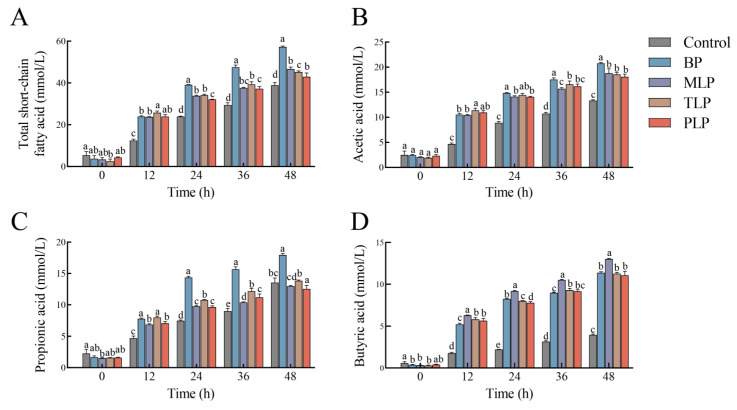
Concentrations of short-chain fatty acids (SCFAs) during in vitro fermentation: (**A**) total SCFAs, (**B**) acetic acid, (**C**) propionic acid and (**D**) butyric acid. Data were expressed as mean ± SD. Different letters indicate significant differences among different polysaccharides at the same time point, *p* < 0.05. The control group (no additional carbon source supplement); BP group (bergamot polysaccharide supplement); MLP (mono-Laoxianghuang polysaccharide supplement); TLP (tri-Laoxianghuang polysaccharide supplement); PLP (penta-Laoxianghuang polysaccharide supplement).

**Figure 5 foods-12-02878-f005:**
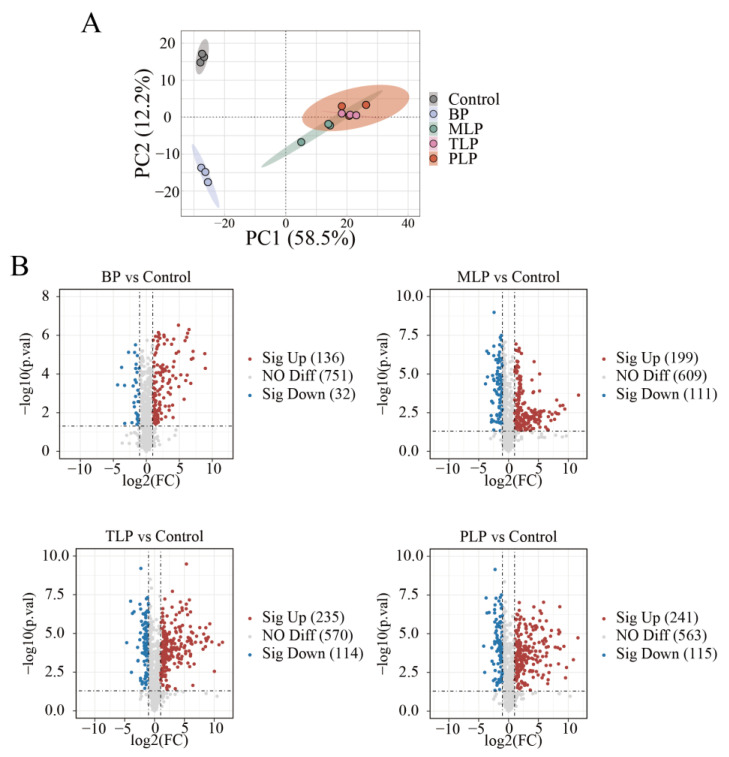
Metabolite change after in vitro fermentation to 48 h. (**A**) Metabolic profile presented by a principal component analysis, (**B**) volcano plots of differentially expressed metabolites compared to the control group for BP, MLP, TLP and PLP groups. Each bubble represents one metabolite, and the color of the bubble represents the differential change: up-regulated, red; down-regulated, blue; not significantly changed, gray. The control group (no additional carbon source supplement); BP (bergamot polysaccharide supplement); MLP (mono-Laoxianghuang polysaccharide supplement); TLP (tri-Laoxianghuang polysaccharide supplement); PLP (penta-Laoxianghuang polysaccharide supplement).

**Figure 6 foods-12-02878-f006:**
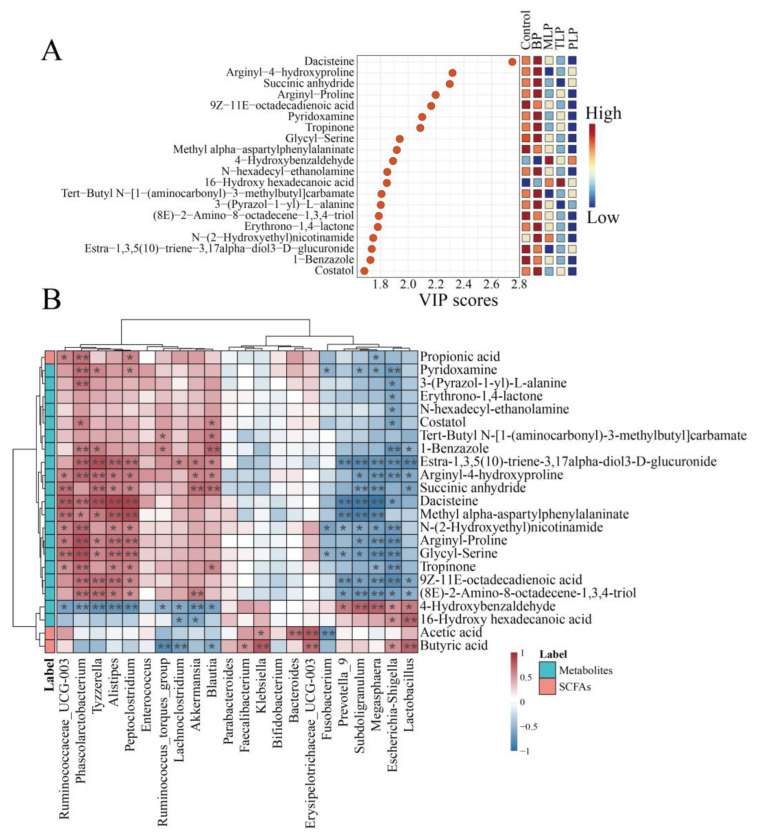
The significant markers screening and correlation analysis. (**A**) Screened 20 significant markers with the highest variable importance in the projection (VIP) scores. (**B**) Spearman correlations between metabolites (the 20 significant markers and short-chain fatty acids) and gut microbiota at the genus level with the highest relative abundance from all the groups, where red and blue cells indicate positive and negative correlations, respectively. * *p* < 0.05, ** *p* < 0.01. The control group (no additional carbon source supplement); BP (bergamot polysaccharide supplement); MLP (mono-Laoxianghuang polysaccharide supplement); TLP (tri-Laoxianghuang polysaccharide supplement); PLP (penta-Laoxianghuang polysaccharide supplement).

## Data Availability

Data is contained within the article or Appendix A.

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
