# Peer review of "Comparative Study on In Vitro Fermentation Characteristics of the Polysaccharides Extracted from Bergamot and Fermented Bergamot"

_foods, 2023, doi:10.3390/foods12152878_

Round 1

Reviewer 1 Report

Current and up-to-date topic. Fermented foods and the benefits of fermentation are on the agenda.

The definition of "prebiotics" given in lines 35-36 is somewhat incomplete. Please complete or explain it further.

The experimental design is not clear. How many bergamot samples? One and three replicate determinations? Or three samples? And were these different, independent samples?

Regarding methodology, why was a Soil DNA extraction kit used and not a Food DNA extraction kit, for example?

Concerning the in vitro fermentation experiment with the polysaccharides, how many samples were taken at each time point?

Why are there some sentences in red along the manuscript?

Author Response

Responses to Reviewer 1

Dear Editors and Reviewers:

Thank you for your kind letter and the reviewers’ comments concerning our manuscript entitled “Comparative study on in vitro fermentation characteristics of the polysaccharides extracted from bergamot and fermented bergamot” (Journal: Foods, Manuscript ID: foods-2466394). Those comments are all valuable and very helpful for revising and improving our paper, as well as the important guiding significance to our research. We have carefully addressed them point-to-point to revise the paper. We have marked all the modifications in the manuscript in RED color for easy recognition by you. You can find our detailed responses in the following sections. We appreciate for Editors/Reviewers’ warm work earnestly and hope that the correction will meet with approval.

Once again, thank you very much for your comments and suggestions. We look forward to hearing from you again.

Best Regards,

Yang Zheng

E-mail: an2zhengyang@126.com

Jie Gao

E-mail: gaojie@gxu.edu.cn

#####################################################################

Response to Reviewer #1:

Comments and Suggestions for Authors

Current and up-to-date topic. Fermented foods and the benefits of fermentation are on the agenda.

Response: Thank you very much for your careful review and your recognition of our work. According to your instructive suggestions, this manuscript has been revised carefully. The followings are our point-by-point responses.

  1. The definition of "prebiotics" given in lines 35-36 is somewhat incomplete. Please complete or explain it further.

Response: Thanks for your suggestion, and now we have supplemented the information about "prebiotics" in Line 34-37.

“Furthermore, certain polysaccharides such as inulin are considered prebiotics, which as a dietary supplement are not digested and absorbed in gastrointestinal tract but are utilized by gut microbiota to promote host health.”

  1. The experimental design is not clear. How many bergamot samples? One and three replicate determinations? Or three samples? And were these different, independent samples?

Response: Sorry for the unclear description. Each polysaccharide was performed in triplicate during in vitro fermentation. Independent samples in the same treatment group were separately experimented and analyzed. And now we have added the information in Line 114-115.

“Each group was performed three independent sample replicates. Three samples from each group were collected separately at 0, 12, 24, 36, and 48 hours after fermentation. They were stored at −80 °C for further study.”

  1. Regarding methodology, why was a Soil DNA extraction kit used and not a Food DNA extraction kit, for example?

Response: We apologize for the misrepresentation of the kit; we have confirmed that HiPure stool DNA kits were used in our work. The gut microbiota in the samples were derived from human stools. Stool DNA kits are widely used in gut microbiota analysis [1-3]. Therefore, stool DNA extraction kits were preferable in our work. And we have revised kits in Line 117.

References:

(1).  Hou, L.; Yu, C.; Zhang, L.; Zhang, F.; Linhardt, R.J.; Chen, S.; Ye, X.; Hou, Z. Structure and microbial-modulating evaluation of sulfhydryl-modified pectin. Food Hydrocolloids. 2023, 108977.

(2).  Sun, Y.; Tang, Z.; Hao, T.; Qiu, Z.; Zhang, B. Simulated Digestion and Fermentation In Vitro by Obese Human Gut Microbiota of Sulforaphane from Broccoli Seeds. Foods. 2022, 11, 4016.

(3).  Lu, X.; Xu, H.; Fang, F.; Liu, J.; Wu, K.; Zhang, Y.; Wu, J.; Gao, J. In vitro effects of two polysaccharide fractions from Laminaria japonica on gut microbiota and metabolome. Food Funct. 2023, 14, 3379-3390.

  1. Concerning the in vitro fermentation experiment with the polysaccharides, how many samples were taken at each time point?

Response: We are thankful for your comment. During the in vitro fermentation experiment, each treatment group was assigned three independent experiments. And three samples were collected once at each time point for further study. Now we have modified the description in Line 114-115.

“Three samples from each group were collected separately at 0, 12, 24, 36, and 48 hours after fermentation. They were stored at −80 °C for further study.”

  1. Why are there some sentences in red along the manuscript?

Response: Thank you very much for your careful review. We have modified red sentences to black that should not appear.

Reviewer 2 Report

The manuscript by Wu and co-workers (Comparative study on in vitro fermentation characteristics of the polysaccharides extracted from bergamot and fermented bergamot) is certainly within the scope of Foods. In general, the manuscript is well written, but some issues need to be addressed:

-L11 (Abstract) and also L92 “Laoxianghuang (fermented 1, 3 and 5 years from bergamot)” needs to be better explained. The bergamot was fermented during 5 years? It is hard to believe that something could be “fermented” during such a long time. Or the bergamot were fermented, and then “stored” for 1, 3, and 5 years…??? From the Discussion (L323-326), it seems that it is “storage period”.

-L75-78 the bergamot were homogenized, and then “form a powder”… there is something missing (drying?).

-L85-88 needs revision. The treatment with “trichloroacetic acid (TCA) to reach a final concentration of 4% (w/v) and placed at 4 ℃ for 10 h.” is repeated twice, makes no sense.

-L105-106 “The monosaccharides composition was determined according to our previous study”…. which study? (see also below)

-L121-122 needs revision (confusing).

-L151 “The process was repeated twice.” needs revision (confusing)

-Figure 1A seems to show exactly the same data that is shown in Table 1 of the group´s previous publication (ref. [11] in the list of references). Figure 1 should be reviewed to avoid duplication of data already published previously.

-Figure 2C is not mentioned in the text.

-Figure 6B, L303, it is mentioned a “***p < 0.001”, but actually this reviewer could not see any “***” in the figure….

Author Response

Responses to Reviewer 2

Dear Editors and Reviewers:

Thank you for your kind letter and the reviewers’ comments concerning our manuscript entitled “Comparative study on in vitro fermentation characteristics of the polysaccharides extracted from bergamot and fermented bergamot” (Journal: Foods, Manuscript ID: foods-2466394). Those comments are all valuable and very helpful for revising and improving our paper, as well as the important guiding significance to our researches. We have carefully addressed them point-to-point to revise the paper. We have marked all the modifications in the manuscript in RED color for easy recognition by you. You can find our detailed responses in the following sections. We appreciate for Editors/Reviewers’ warm work earnestly, and hope that the correction will meet with approval.

Once again, thank you very much for your comments and suggestions. We look forward to hearing from you again.

Best Regards,

Yang Zheng

E-mail: an2zhengyang@126.com

Jie Gao

E-mail: gaojie@gxu.edu.cn

#####################################################################

Response to Reviewer #2:

Comments to the Author

The manuscript by Wu and co-workers (Comparative study on in vitro fermentation characteristics of the polysaccharides extracted from bergamot and fermented bergamot) is certainly within the scope of Foods. In general, the manuscript is well written, but some issues need to be addressed:

Response: Thank you very much for your careful review. We are grateful for your recognition of this work. According to your instructive comments, the followings are our point-by-point responses.

  1. -L11 (Abstract) and also L92 “Laoxianghuang (fermented 1, 3 and 5 years from bergamot)” needs to be better explained. The bergamot was fermented during 5 years? It is hard to believe that something could be “fermented” during such a long time. Or the bergamot were fermented, and then “stored” for 1, 3, and 5 years…??? From the Discussion (L323-326), it seems that it is “storage period”.

Response: Many thanks for your comment. The production way of Laoxianghuang is a traditional food processing technology in Chaoshan, Guangdong Province, China. It is a product from bergamot through salting, desalting, sugaring, cooking, drying, and fermentation [1]. Fermentation refers to the pickle and storage for different time periods (1, 3, and 5 years) [2]. After fermentation, the Laoxianghuang shows black and soft characteristics (Fig. 1 in the attached word) [3]. In order to explain clearly the information, now we have modified the expression in Line 46-47.

References:

(1). Yaqun, L.; Hanxu, L.; Wanling, L.; Yingzhu, X.; Mouquan, L.; Yuzhong, Z.; Lei, H.; Yingkai, Y.; Yidong, C. SPME-GC-MS combined with chemometrics to assess the impact of fermentation time on the components, flavor, and function of Laoxianghuang. Front Nutr. 2022, 9, 915776.

(2). Zheng, Y.Z.; Guo, S.J.; Yang, Y.L.; Gu, X.X.; Li, X.T.; Zhuang, X.D.; Lin, Z.Z.; Zhang, Z.X. roduction Process of Preserved Fruits Laoxianghuang. Academic Periodical of Farm Products Processing. 2014, 340, 44-45.

(3). Yang, D.; Chen, X.A.; Yang, Y.J.; Cai, S.; Chen, S.X.; Zhou, A.M. Using E-nose, HS-GC-IMS, and HS-SPME-GC-MS to Differentiate the Volatile Components of Lao Xianghuang Fermented for Different Years. MODERN FOOD SCIENCE & TECHNOLOGY. 2021, 38, 313-323.

  1. -L75-78 the bergamot were homogenized, and then “form a powder”… there is something missing (drying?).

Response: Sorry for the unclear statement. We have supplemented the expression in Line 77-78.

“After drying at 40 ℃, ...”

  1. -L85-88 needs revision. The treatment with “trichloroacetic acid (TCA) to reach a final concentration of 4% (w/v) and placed at 4 ℃ for 10 h.” is repeated twice, makes no sense.

Response: Many thanks for the mentioned error. We have removed duplicate meaningless descriptions in Line 87

  1. -L105-106 “The monosaccharides composition was determined according to our previous study”…. which study? (see also below)

Response: Thank you for your careful review. Citation was updated in Line 104.

  1. -L121-122 needs revision (confusing).

Response: Thanks for pointing out the confusing expression. And now we have revised the sentence in Line 121-122.

“Raw reads were further filtered to get quality reads by using FASTP (version 0.18.0).”

  1. -L151 “The process was repeated twice.” needs revision (confusing)

Response: Thanks for pointing out the confusing expression. The expression has been revised in Line 150-152.

“The step regarding methanol extraction and centrifugation was repeated twice. Subsequently, the supernatant of two centrifugation was mixed and used for high-performance liquid chromatography−tandem mass spectrometry (HPLC−MS/MS) analysis.”

  1. -Figure 1A seems to show exactly the same data that is shown in Table 1 of the group´s previous publication (ref. [11] in the list of references). Figure 1 should be reviewed to avoid duplication of data already published previously.

Response: Thanks for your suggestion, the information of monosaccharide composition has been moved to Figure S1 at supplementary material. And Figure 1 has been revised in Line 197-199.

  1. -Figure 2C is not mentioned in the text.

Response: Thank you very much for your review, now we have added “Figure 2C”, as shown in Line 191.

  1. -Figure 6B, L303, it is mentioned a “***p < 0.001”, but actually this reviewer could not see any “***” in the figure….

Response: Thanks for your suggestion, and now we have removed the redundant expression in Line 300.

Round 2

Reviewer 2 Report

The authors addressed all issues raised by this reviewer.